# Minimax Rates of *ℓ*_*p*_-Losses for High-Dimensional Linear Errors-in-Variables Models over *ℓ*_*q*_-Balls

**DOI:** 10.3390/e23060722

**Published:** 2021-06-05

**Authors:** Xin Li, Dongya Wu

**Affiliations:** 1School of Mathematics, Northwest University, Xi’an 710069, China; lixin@nwu.edu.cn; 2School of Information Science and Technology, Northwest University, Xi’an 710069, China

**Keywords:** sparse linear regression, errors-in-variables model, minimax rate, Kullback–Leibler divergence, information-theoretic limitations

## Abstract

In this paper, the high-dimensional linear regression model is considered, where the covariates are measured with additive noise. Different from most of the other methods, which are based on the assumption that the true covariates are fully obtained, results in this paper only require that the corrupted covariate matrix is observed. Then, by the application of information theory, the minimax rates of convergence for estimation are investigated in terms of the $\ell_{p}\;\left(1\leq p<\infty \right)$-losses under the general sparsity assumption on the underlying regression parameter and some regularity conditions on the observed covariate matrix. The established lower and upper bounds on minimax risks agree up to constant factors when p=2, which together provide the information-theoretic limits of estimating a sparse vector in the high-dimensional linear errors-in-variables model. An estimator for the underlying parameter is also proposed and shown to be minimax optimal in the $\ell_{2}$-loss.

## 1. Introduction

In various fields of applied sciences and engineering, such as machine learning [1], a fundamental problem is to estimate an underlying parameter $\beta^{*}\in R^{d}$ of a linear regression model as follows
(1)$$y_{i}=\langle X_{i\cdot },\beta^{*}\rangle +e_{i},\;\mathrm{for}\;i=1,2,\ldots ,n,$$
where ${\left\{\left(X_{i\cdot },y_{i}\right)\right\}}_{i=1}^{n}$ are i.i.d. observations ($X_{i\cdot }\in R^{d}$) and $e\in R^{n}$ is the random noise. In matrix form, model (Equation 1) can be written as $y=X\beta^{*}+e$, where $X={\left(X_{1\cdot },\ldots ,X_{n\cdot }\right)}^{⊤}\in R^{n\times d}$ and $y,e\in R^{n}$. The covariates $X_{i\cdot }\;\left(i=1,2,\ldots ,n\right)$ are always assumed to be fully observed in standard formulations. However, this assumption is far away from reality since, in general, the measurement error cannot be avoided. In many real-world applications, due to the lack of observation or the instrumental constraint, the collected data, such as remote sensing data, may always be perturbed and tend to be noisy [2]. It has been shown in [3] that misleading inference results may be obtained if the method for clean data is applied to the noisy data naively. Therefore, it is more realistic to explore the case where only the corrupted covariates of the corresponding true covariates $X_{i\cdot }$’s are obtained; see, e.g., [4]. This is known as the measurement error model in the literature.

Estimation in the presence of measurement errors has attracted a lot of interest for a long time. Bickel and Ritov [5] first studied the linear measurement error models and proposed an efficient estimator. Then, Stefanski and Carroll [6] investigated the generalized linear measurement error models and constructed consistent estimators. Extensive results have also been established on parameter estimation and variable selection for both parametric or nonparametric settings; see [7,8] and references therein. It should be noted that these results are only applicable to classical low-dimensional (i.e., n≥d) statistical models.

In the past two decades, high-dimensional statistical models, where the number of observations is much less than the number of predictors (i.e., n≪d), have been paid much attention and have achieved fruitful results in a wide range of research areas; see [9,10] for a detailed review. Most of the existing results are only suitable for models with clean data, while some researchers have began to focus on the measurement error case. For example, Loh and Wainwright studied the high-dimensional sparse linear regression model with corrupted covariates. Though the proposed estimator involves solving a nonconvex optimization problem, they proved that the global and stationary points are statistically consistent; see [11,12]. Datta and Zou [13] proposed the Convex Conditioned Lasso (CoCoLasso), which enjoys the convex benefit of the Lasso in both estimation and algorithm and can handle a class of corrupted datasets, including the cases of additive or multiplicative measurement error. Li et al. [14] investigated a general nonconvex estimation method from statistical and computational aspects and the results can be immediately applied to the corrupted errors-in-variables linear regression.

Apart from the study on statistical convergence rates and designing efficient algorithms to solve certain estimators, it is also fundamental to information-theoretic limitations of statistical inference to understand the computationally efficient procedures. Such fundamental limits are usually studied by virtue of the minimax rates, which aim to find an estimator that minimizes the worst-case loss and, thus, can reveal gaps between the performance of some computationally efficient algorithm and that of an optimal algorithm. The minimax rate is always analyzed from two aspects, namely the informational lower bounds and statistical upper bounds. In the information-theoretic aspect, the Kullback–Leibler (KL) divergence is always used to provided lower bounds [15]. Recently, in [16], Loh provides a detailed review of a variety of techniques utilized to derive information-theoretic lower bounds for minimax estimation and learning, focusing on the problem settings with community recovery, parameter and function estimation, and online learning for multi-armed bandits. In the statistical aspect, a special estimator is always constructed to derive upper bounds; see, e.g., [17,18]. For the high-dimensional linear regression with additive errors, Loh and Wainwright [19] established minimax rates of convergence for estimating the unknown parameter in the $\ell_{2}$-loss. The proposed estimator was also shown to be minimax optimal in the additive error case under the $\ell_{2}$-loss, assuming that the true parameter is exact sparse, that is, $\beta^{*}$ has at most s≪d nonzero elements, which is also known as the exact sparsity assumption.

However, this exact assumption may be sometimes too restrictive to be satisfied in some real applications. For example, in the field of image processing, it is a standard phenomenon that wavelet coefficients for images usually exhibit an exponential decay, but do not need to be almost 0 (see, e.g., [20]). Other applications under high-dimensional scenarios include compressed sensing [21], genomic analysis [22], signal processing [23], and so on, where it is not suitable to impose an exact sparsity assumption on the underlying parameter. Hence, it is necessary to investigate minimax rates of estimation when the exact sparse assumption does not hold.

Our main purpose in the present study is to investigate the more general situation that coefficients of the true parameter are not almost zeros and then provide minimax rates of convergence for estimation in sparse linear regression with additive errors. More precisely speaking, we assume that for q∈[0,1] fixed, the $\ell_{q}$-norm of $\beta^{*}$ defined as $∥\beta^{*}{∥}_{q}:=(\sum_{j=1}^{p}|\beta_{j}^{*}{{|}^{q})}^{1/q}$ is bounded from above. Note that this assumption is reduced to the exact sparsity assumption when q=0. When q∈(0,1], this type of sparsity is known as the soft sparsity. The exact sparsity assumption has been widely used for statistical inference, while the soft sparsity assumption attracts relatively little attention apart from the work [24,25,26]. Specifically, under both exact and soft sparsity assumptions, Raskutti et al. [24] and Ye and Zhang [26] provided minimax rates of convrgence for estimation in high-dimensional linear regression, respectively; Wang et al. [25] developed the optimal rates of convergence and proposed an adaptive $\ell_{q}$-aggregation strategy via model mixing which attains the established optimal rate automatically. It is worth noting that results in [24,25,26] are all obtained for clean data and cannot be applied to the errors-in-variables model. This is a fundamental difference from our present study.

The main contributions of this paper are as follows. By assuming that the regression parameter is of soft sparsity, in the information-theoretic aspect we establish lower bounds on the minimax risks for $\ell_{p}\;\left(1\leq p<\infty \right)$-losses by virtue of the mutual information which hold for any arbitrary estimator for the model regardless of the specific method. In the statistical aspect, we propose an estimator which can be solved efficiently and then provide upper bounds on the $\ell_{2}$-loss between the estimator and the true parameter. Moreover, the lower and upper bounds when p=2 agree up to constant factors, implying that proposed estimator is minimax optimal in the $\ell_{2}$-loss.

The remainder of this paper is organized as follows. In Section 2, we provide background on the errors-in-variables linear regression model and some regularity conditions on the observed covariate matrix. In Section 3, we establish our main results on lower and upper bounds on minimax risks for $\ell_{p}\;\left(1\leq p<\infty \right)$-losses over $\ell_{q}$-balls. Conclusions and future work are discussed in Section 4.

We end this section by introducing some notations for future reference. We use Greek lowercase letter β to denote the vectors. All vectors are column vectors following classical mathematical convention. A vector β is supported on *S* if and only if $S=\{i\in \left\{1,2,\cdots ,d\right\}:\beta_{i}\neq 0\}$, and *S* is the support of β denoted by supp(β), namely supp(β)=S. For d≥1, let $I_{d}$ stand for the d×d identity matrix. For a matrix $X\in R^{n\times d}$, let $X_{ij}\;\left(i=1,\cdots ,n,j=1,2,\ldots ,d\right)$ denote its ij-th entry, $X_{i\cdot }\;\left(i=1,\cdots ,n\right)$ denote its *i*-th row, $X_{\cdot j}\;\left(j=1,2,\ldots ,d\right)$ denote its *j*-th column.

## 2. Problem Setup

In this section, we begin with a precise formulation of the problem and then impose some regularity assumptions on the observed matrix.

Recall the standard linear regression model (Equation 1). One of the main types of measurement errors is the additive error. Specifically, for each i=1,2,…,n, we observe $Z_{i\cdot }=X_{i\cdot }+W_{i\cdot }$, where $W_{i\cdot }\in R^{d}$ is a random vector independent of $X_{i\cdot }$ with mean 0 and known covariance matrix $\Sigma_{w}$. When the noise covariance $\Sigma_{w}$ is unknown, there are some method to estimate it from the observed data; see, e.g., [4]. For example, a simple method is to estimate $\Sigma_{w}$ from blank independent observations of the noise. Specifically, suppose that one independently observes a matrix $W_{0}\in R^{n\times d}$ with *n* i.i.d. vectors of noise. Then we use $\Sigma_{w}=\frac{1}{n}W_{0}^{⊤}W_{0}$ as the estimate of $\Sigma_{w}$. Some other sophisticated variant of this method in are also provided in [4].

Throughout this paper, we assume that for i=1,2,…,n, the vectors $X_{i\cdot }$, $W_{i\cdot }$, and *e* are Gaussian with mean 0 and covariance matrices $\sigma_{x}^{2}I_{d}$$\left(\sigma_{x}>0\right)$, $\sigma_{w}^{2}I_{d}$, and $\sigma_{e}^{2}I_{n}$, respectively, and we write $\sigma_{z}^{2}=\sigma_{x}^{2}+\sigma_{w}^{2}$ for simplicity.

According to the previous works of [11,12], we fix i∈{1,2,…,n} and write $\Sigma_{x}$ to denote the covariance matrix of $X_{i\cdot }$ (i.e., $\Sigma_{x}=\mathrm{cov}\left(X_{i\cdot }\right)=\sigma_{x}^{2}I_{d}$). Let $\left(\hat{\Gamma },\hat{\Upsilon }\right)$ stand for the estimators for $\left(\Sigma_{x},\Sigma_{x}\beta^{*}\right)$ which only depend on the observed data ${\left\{\left(Z_{i\cdot },y_{i}\right)\right\}}_{i=1}^{n}$. As has been discussed in [11], an unbiased and suitable choice of the surrogate pair $\left(\hat{\Gamma },\hat{\Upsilon }\right)$ for the additive error case is given by
$$\hat{\Gamma }:=\frac{Z^{⊤}Z}{n}-\Sigma_{w}\;and\;\hat{\Upsilon }:=\frac{Z^{⊤}y}{n}.$$

Under the high-dimensional scenario (n≪d), the matrix $\hat{\Gamma }$, which is the estimator of $\Sigma_{x}$ in the corrupted case, is always negative definite. To be specific, the matrix $Z^{⊤}Z$ has rank at most *n*, and then the positive definite matrices $\Sigma_{w}$ are subtracted to obtain $\hat{\Gamma }$. Consequently, $\hat{\Gamma }$ cannot be guaranteed to be positive definite regardless of the amount of noise. However, this does not affect the current result. Particularly, though the negative definiteness of $\hat{\Gamma }$ leads to a nonconvex optimization problem in estimating $\beta^{*}$ (cf. (Equation 14)) as well as the upper bound, a weaker condition (cf. Assumption 2) allows further analysis.

Instead of assuming the regression parameter $\beta^{*}$ is exact sparse (i.e., supp(β)≪d), we use a general notion to characterize the sparsity of $\beta^{*}$. Specifically, we assume that for q∈[0,1], and a radius $R_{q}>0$, $\beta^{*}\in B_{q}\left(R_{q}\right)$, where
$$B_{q}\left(R_{q}\right):=\{\beta \in R^{d}{:||\beta ||}_{q}^{q}=\sum_{j=1}^{d}|\beta_{j}{|}^{q}\leq R_{q}\}.$$

The use of $\ell_{q}$-ball is a common and popular way to measure the degree of sparsity (accurately, the above sets are not real “balls”, as they fail to be convex when q∈[0,1)). Note that $\beta \in B_{0}\left(R_{0}\right)$ corresponds to the case that β is exact sparse, while for q∈(0,1], $\beta \in B_{q}\left(R_{q}\right)$ corresponds to the case of weak sparsity, which endows a certain decay rate on the ordered entries of β. Throughout this paper, let q∈[0,1] be fixed, and we assume that $\beta^{*}\in B_{q}\left(R_{q}\right)$ unless otherwise specified. Moreover, without loss of generality, we assume that $∥\beta^{*}{∥}_{2}=1$ and define $S_{2}\left(1\right):=\{\beta \in R^{d}{\;|\;∥\beta ∥}_{2}=1\}$, i.e., the $\ell_{2}$ unit sphere. Then it follows that $\beta^{*}\in B_{q}\left(R_{q}\right)\cap S_{2}\left(1\right)$.

In order to estimate the regression parameter, one usually considers an estimator $\hat{\beta }:R^{n\times d}\times R^{n}\to R^{d}$, which is a measurable function of the observed data ${\left\{\left(Z_{i\cdot },y_{i}\right)\right\}}_{i=1}^{n}$. Then, for the purpose of assessing the estimation quality of $\hat{\beta }$, it is typical to introduce a loss function $L(\hat{\beta },\beta^{*})$, which represents the loss incurred by the estimator $\hat{\beta }$ when the true parameter $\beta^{*}\in B_{q}\left(R_{q}\right)\cap S_{2}\left(1\right)$. Finally, in the minimax formalism, we aim to choose an estimator that minimizes the following worst-case loss
$$\min_{\hat{\beta }}\max_{\beta^{*}\in B_{q}\left(R_{q}\right)\cap S_{2}\left(1\right)}L\left(\hat{\beta },\beta^{*}\right).$$

Specifically, in this paper, we shall consider the $\ell_{p}$-losses for p∈[1,+∞) as follows
$$L_{p}\left(\hat{\beta },\beta^{*}\right):={∥\hat{\beta }-\beta^{*}∥}_{p}^{p}.$$

We then impose some regularity conditions on the observed matrix *Z*, which are beneficial to analyze the minimax rates. The first assumption requires that the columns of *Z* are bounded from above in $\ell_{2}$-norm.

**Assumption** **1**(Column normalization)**.**
*There exists a constant $0<\kappa_{c}<+\infty$ such that*
$$\frac{1}{\sqrt{n}}\max_{j=1,2,\ldots ,d}{∥Z_{\cdot j}∥}_{2}\leq \kappa_{c}.$$

The second assumption imposes a lower bound on the restricted eigenvalue of the surrogate gram matrix $\hat{\Gamma }$, which in other words is a lower bound for the restricted curvature.

**Assumption** **2**(Restricted eigenvalue condition)**.**
*There exists a constant $\kappa_{l}>0$ and a function $\tau_{l}\left(n,d\right)$ such that for all $\beta \in B_{q}\left(2R_{q}\right)$,*
$$\beta^{⊤}\hat{\Gamma }\beta \geq \kappa_{l}{∥\beta ∥}_{2}^{2}-\tau_{l}\left(n,d\right).$$

**Remark 1.** (i) *Note that though we focused on the random design case in this article, Assumptions 1 and 2 are stated in deterministic form. This choice is to make them universal to both fixed and random design matrices. Specifically, previous studies have shown that Assumptions 1 and 2 can be satisfied by a wide range of random matrices with high probability; see, e.g., [11,14,27]. Meanwhile, Assumptions 1 and 2 provide the possibility to analyze the fixed design case in which the matrices are usually chosen by researchers with suitable constants, i.e., $\kappa_{c}$ in Assumption 1 and $\kappa_{l},\tau_{l}$ in Assumption 2. This deterministic form of the regularity condition on the design matrix is also adopted in the field of modern high-dimensional statistics and machine learning; see, e.g., [11,12,14,28].*(ii) *For the Gaussian model we assumed that for i=1,2,…,n, the vectors $X_{i\cdot }$ and $W_{i\cdot }$ are independently Gaussian with mean 0 and covariance matrices $\sigma_{x}^{2}I_{d}$ and $\sigma_{w}^{2}I_{d}$, respectively, and the observed covariate $Z_{i\cdot }$ is also Gaussian with mean 0 and covariance matrix $\Sigma_{z}=\left(\sigma_{x}^{2}+\sigma_{w}^{2}\right)I_{d}$. Recall that $\sigma_{z}^{2}=\sigma_{x}^{2}+\sigma_{w}^{2}$, then one has that $Z_{i\cdot }∼N\left(0,\sigma_{z}^{2}I_{d}\right)$. Furthermore, since the observations are i.i.d., each column $Z_{\cdot j}\;\left(j=1,\ldots ,d\right)$ has i.i.d. elements, and thus $∥Z_{\cdot j}{∥}_{2}^{2}\;\left(j=1,\ldots ,d\right)$ obeys the $\chi^{2}$ distribution with freedom n. Then, for Assumption 1, it follows immediately from [27] [Appendix I] on standard tail bounds for $\chi^{2}$-variates and union bounds that there exist universal positive constants $\left(c_{0},c_{1}\right)$ and that Assumption 1 holds with $\kappa_{c}=c_{0}\sigma_{z}\sqrt{\frac{\log d}{n}}$ with probability at least $1-c_{1}\exp\left(-logd\right)$.*(iii) *As for Assumption 2, it follows from [29] [Lemma 1] that there exist universal positive constants $\left(c_{0},c_{1},c_{2}\right)$, such that Assumption 2 holds with $\kappa_{l}=\frac{\sigma_{x}^{2}}{2}$ and $\tau_{l}=c_{0}\sigma_{x}^{2}\max\left(\frac{\sigma_{z}^{4}}{\sigma_{x}^{4}},1\right)\frac{\log d}{n}$ with probability at least $1-c_{1}\exp\left(-c_{2}n\min\left(\frac{\sigma_{x}^{4}}{\sigma_{z}^{4}},1\right)\right)$.*

## 3. Main Results

In this section, we turn to our main results on lower and upper bounds on minimax risks. We first begin with deriving intermediate results under Assumptions 1 and 2, then we will turn to our main results on probabilistic consequences by virtue of Remark 1(ii), (iii) on the conditions to guarantee Assumptions 1 and 2.

Let $P_{\beta }$ denote the distribution of *y* in the linear regression model with additive errors, when β is given and *Z* is observed. The following lemma tells us the KL divergence between the distributions induced by two different parameters $\beta ,\beta^{\prime }\in B_{q}\left(R_{q}\right)$. The KL divergence plays a key role in establishing the information-theoretic related lower bound. Recall that for two distributions P and Q which have densities dP and dQ with respect to some base measure μ, the KL divergence is defined by $D\left(P||Q\right)=\int \log\frac{dP}{dQ}P\left(d\mu \right)$.

**Lemma** **1.**
*In the additive error setting, the KL divergence between the distributions induced by any $\beta ,\beta^{\prime }\in B_{q}\left(R_{q}\right)\cap S_{2}\left(1\right)$ is equal to*
$$D(P_{\beta }\left|\right|P_{\beta^{\prime }})=\frac{\sigma_{x}^{4}}{2\sigma_{z}^{2}\left(\sigma_{x}^{2}\sigma_{w}^{2}+\sigma_{z}^{2}\sigma_{e}^{2}\right)}{∥Z\left(\beta -\beta^{\prime }\right)∥}_{2}^{2}.$$


**Proof.** For each i=1,2,…,n fixed, by the model setting, $\left(y_{i},Z_{i\cdot }\right)$ is jointly Gaussian with mean **0**. Then by some elementary algebra to compute the covariances, one has that
$$\left[\begin{array}{c} y_{i}\\ Z_{i\cdot } \end{array}\right]∼N\left(\left[\begin{array}{c} 0\\ 0 \end{array}\right],\left[\begin{array}{cc} \beta^{⊤}\Sigma_{x}\beta +\sigma_{e}^{2} & \beta^{⊤}\Sigma_{x}\\ \Sigma_{x}\beta & \Sigma_{x}+\Sigma_{w} \end{array}\right]\right).$$Then, it follows from standard results on the conditional distribution of Gaussian variables that
(2)$$y_{i}|Z_{i\cdot }∼N\left(\beta^{⊤}\Sigma_{x}\Sigma_{z}^{-1}Z_{i\cdot },\beta^{⊤}\left(\Sigma_{x}-\Sigma_{x}\Sigma_{z}^{-1}\Sigma_{x}\right)\beta +\sigma_{e}^{2}\right).$$Now assume that $\sigma_{e}$ and $\sigma_{w}$ are not both 0; otherwise, the conclusion holds trivially. Since $P_{\beta }$ is a product distribution of $y_{i}|Z_{i\cdot }$ over all i=1,2,…,n, it follows from (Equation 2) that
(3)$$\begin{array}{cc} D(P_{\beta }||P_{\beta^{\prime }}) & =E_{P_{\beta }}\left[\log\frac{P_{\beta }\left(y\right)}{P_{\beta^{\prime }}\left(y\right)}\right]\\ \; & =E_{P_{\beta }}\left[\frac{n}{2}\log\left(\frac{\sigma_{\beta^{\prime }}^{2}}{\sigma_{\beta }^{2}}\right)-\frac{∥y-Z\Sigma_{z}^{-1}\Sigma_{x}{\beta ∥}_{2}^{2}}{2\sigma_{\beta }^{2}}+\frac{∥y-Z\Sigma_{z}^{-1}\Sigma_{x}\beta^{\prime }{∥}_{2}^{2}}{2\sigma_{\beta^{\prime }}^{2}}\right]\\ \; & =\frac{n}{2}\log\left(\frac{\sigma_{\beta^{\prime }}^{2}}{\sigma_{\beta }^{2}}\right)+\frac{n}{2}\left(\frac{\sigma_{\beta }^{2}}{\sigma_{\beta^{\prime }}^{2}}-1\right)+\frac{1}{2\sigma_{\beta^{\prime }}^{2}}{∥Z\Sigma_{z}^{-1}\Sigma_{x}\left(\beta -\beta^{\prime }\right)∥}_{2}^{2}, \end{array}$$
where $\sigma_{\beta }^{2}:=\beta^{⊤}\left(\Sigma_{x}-\Sigma_{x}\Sigma_{z}^{-1}\Sigma_{x}\right)\beta +\sigma_{e}^{2}$, and $\sigma_{\beta^{\prime }}^{2}$ is given analogously. Since $\Sigma_{x}=\sigma_{x}^{2}I_{n}$, $\Sigma_{w}=\sigma_{w}^{2}I_{n}$, and ${∥\beta ∥}_{2}=1$ by the assumptions, we immediately arrive at that
$$\sigma_{\beta }^{2}=\left(\sigma_{x}^{2}-\frac{\sigma_{x}^{4}}{\sigma_{z}^{2}}\right){∥\beta ∥}_{2}^{2}+\sigma_{e}^{2}=\frac{\sigma_{x}^{2}\sigma_{w}^{2}}{\sigma_{z}^{2}}+\sigma_{e}^{2}.$$Substituting this equality into (Equation 3) yields that
$$D(P_{\beta }\left|\right|P_{\beta^{\prime }})=\frac{\sigma_{x}^{4}}{2\sigma_{z}^{2}\left(\sigma_{x}^{2}\sigma_{w}^{2}+\sigma_{z}^{2}\sigma_{e}^{2}\right)}{∥Z\left(\beta -\beta^{\prime }\right)∥}_{2}^{2}.$$The proof is completed. □

**Proposition** **1.**
*In the additive error setting, suppose that the observed matrix Z satisfies Assumption 1 with $0<\kappa_{c}<+\infty$. Then, for any p∈[1,+∞), there exists a constant $c_{q,p}$ depending only on q and p such that with probability at least 1/2, the minimax $\ell_{p}$-loss over the $\ell_{q}$-ball is lower bounded as*
(4)$$\min_{\hat{\beta }}\max_{\beta^{*}\in B_{q}\left(R_{q}\right)\cap S_{2}\left(1\right)}{∥\hat{\beta }-\beta^{*}∥}_{p}^{p}\geq c_{q,p}{\left[\frac{\sigma_{z}^{2}\left(\sigma_{x}^{2}\sigma_{w}^{2}+\sigma_{z}^{2}\sigma_{e}^{2}\right)}{\sigma_{x}^{4}\kappa_{c}^{2}}\right]}^{\frac{p-q}{2}}R_{q}{\left(\frac{\log d}{n}\right)}^{\frac{p-q}{2}}.$$


**Proof.** For positive numbers δ>0 and ϵ>0, let $M_{p}\left(\delta \right)$ denote the cardinality of a maximal δ-packing of the ball $B_{q}\left(R_{q}\right)$ in the $l_{p}$ metric with elements $\left\{\beta^{1},\beta^{2},\ldots ,\beta^{M}\right\}$, and $N_{2}\left(\epsilon \right)$ denote the minimal cardinality of an ϵ-covering of $B_{q}\left(R_{q}\right)$ in $\ell_{2}$-norm. We follow the standard technique in [30] to transform the estimation on lower bound into a multi-way hypothesis testing problem as follows
(5)$$P\left(\min_{\hat{\beta }}\max_{\beta^{*}\in B_{q}\left(R_{q}\right)\cap S_{2}\left(1\right)}{∥\hat{\beta }-\beta^{*}∥}_{p}^{p}\geq \frac{1}{2^{p}}\delta^{p}\right)\geq \min_{\tilde{\beta }}P\left(B\neq \tilde{\beta }\right),$$
where $B\in R^{d}$ is a random variable uniformly distributed over the packing set $\left\{\beta^{1},\beta^{2},\ldots ,\beta^{M}\right\}$, and $\tilde{\beta }$ is an estimator taking values in the packing set. It then follows from Fano’s inequality [30] that
(6)$$P\left(B\neq \tilde{\beta }\right)\geq 1-\frac{I(y;B)+\log2}{\log M_{p}\left(\delta \right)},$$
where I(y;B) is the mutual information between the random variable *B* and the observation vector $y\in R^{n}$. It now remains to upper bound the mutual information I(y;B). Based on the procedure of [30], the mutual information is upper bounded as
(7)$$I\left(y;B\right)\leq \log N_{2}\left(\epsilon \right)+D(P_{\beta }\left|\right|P_{\beta^{\prime }}).$$Let ${\mathrm{absconv}}_{q}\left(Z/\sqrt{n}\right)$ denote the *q*-convex hull of the rescaled columns of the observed matrix *Z*, that is,
$${\mathrm{absconv}}_{q}\left(Z/\sqrt{n}\right):=\left\{\frac{1}{\sqrt{n}}\sum_{j=1}^{n}\theta_{j}Z_{\cdot j}|\theta \in B_{q}\left(R_{q}\right)\right\},$$
where the normalization factor $1/\sqrt{n}$ is used for convenience. Since *Z* satisfies Assumption 1 [31], [Lemma 4] is applicable to concluding that there exists a set $\left\{Z{\tilde{\beta }}^{1},Z{\tilde{\beta }}^{2},\ldots ,Z{\tilde{\beta }}^{N}\right\}$ such that for all $Z\beta \in {\mathrm{absconv}}_{q}\left(Z\right)$, there exists some index *i* and some constant c>0 such that $∥Z\left(\beta -{\tilde{\beta }}^{i}\right){∥}_{2}/\sqrt{n}\leq c\kappa_{c}\epsilon$. Combining this inequality with Lemma 1 and (Equation 7), one has that the mutual information is upper bounded as
$$I\left(y;B\right)\leq \log N_{2}\left(\epsilon \right)+\frac{\sigma_{x}^{4}}{\sigma_{z}^{2}\left(\sigma_{x}^{2}\sigma_{w}^{2}+\sigma_{z}^{2}\sigma_{e}^{2}\right)}nc^{2}\kappa_{c}^{2}\epsilon^{2}.$$Thus, we obtain by (Equation 6) that
(8)$$P\left(B\neq \tilde{\beta }\right)\geq 1-\frac{\log N_{2}\left(\epsilon \right)+\frac{\sigma_{x}^{4}}{\sigma_{z}^{2}\left(\sigma_{x}^{2}\sigma_{w}^{2}+\sigma_{z}^{2}\sigma_{e}^{2}\right)}nc^{2}\kappa_{c}^{2}\epsilon^{2}+\log2}{\log M_{p}\left(\delta \right)}.$$It remains to choose the packing and covering set radii (i.e., δ and ϵ, respectively) such that (Equation 8) is strictly above zero, say bounded below by 1/2. For the sake of simplicity, denote $\sigma^{2}:=\frac{\sigma_{z}^{2}\left(\sigma_{x}^{2}\sigma_{w}^{2}+\sigma_{z}^{2}\sigma_{e}^{2}\right)}{\sigma_{x}^{4}}$. Suppose that we choose the pair (δ,ϵ) such that
(9a)$$\begin{array}{cc} \frac{c^{2}n}{\sigma^{2}}\kappa_{c}^{2}\epsilon^{2} & \leq \log N_{2}\left(\epsilon \right),\;\mathrm{and} \end{array}$$
(9b)$$\begin{array}{cc} \log M_{p}\left(\delta \right) & \geq 6\log N_{2}\left(\epsilon \right). \end{array}$$As long as $N_{2}\left(\epsilon \right)\geq 2$, it is guaranteed that
(10)$$P\left(B\neq \tilde{\beta }\right)\geq 1-\frac{2\log N_{2}\left(\epsilon \right)+\log2}{6\log N_{2}\left(\epsilon \right)}\geq \frac{1}{2},$$
as desired. It remains to determine the values of the pair (δ,ϵ) satisfying (9). By [31] [Lemma 3], we know that if $\frac{c^{2}n}{\sigma^{2}}\kappa_{c}^{2}\epsilon^{2}=L_{q,2}\left[R_{q}^{\frac{2}{2-q}}{\left(\frac{1}{\epsilon }\right)}^{\frac{2q}{2-q}}\log d\right]$ for some constant $L_{q,2}$ depending only on *q*, then ([Disp-formula FD9a-entropy-23-00722]) is satisfied. Thus, we can choose ϵ satisfying
(11)$$\epsilon^{\frac{4}{2-q}}=L_{q,2}R_{q}^{\frac{2}{2-q}}\frac{\sigma^{2}}{c^{2}\kappa_{c}^{2}}\frac{\log d}{n}.$$In addition, it follows from [31] [Lemma 3] that if δ is chosen to satisfy
(12)$$U_{q,p}\left[R_{q}^{\frac{p}{p-q}}{\left(\frac{1}{\delta }\right)}^{\frac{pq}{p-q}}\log d\right]\geq 6L_{q,2}\left[R_{q}^{\frac{2}{2-q}}{\left(\frac{1}{\epsilon }\right)}^{\frac{2q}{2-q}}\log d\right],$$
for some constant $U_{q,p}$ depending only on *q* and *p*, then (9b) holds. Combining (Equation 11) and (Equation 12), one has that
$$\delta^{p}\leq {\left[\frac{U_{q,p}}{6L_{q,2}}\right]}^{\frac{p-q}{q}}{\left(\epsilon^{\frac{4}{2-q}}\right)}^{\frac{p-q}{2}}R_{q}^{\frac{2-p}{2-q}}=L_{q,2}^{\frac{p-q}{2}}{\left[\frac{U_{q,p}}{6L_{q,2}}\right]}^{\frac{p-q}{q}}R_{q}{\left[\frac{\sigma^{2}}{c^{2}\kappa_{c}^{2}}\frac{\log d}{n}\right]}^{\frac{p-q}{2}}.$$Combining this inequality with (Equation 10) and (Equation 5), we obtain that there exists a constant $c_{q,p}$ depending only on *q* and *p* such that
$$P\left(\min_{\hat{\beta }}\max_{\beta^{*}\in B_{q}\left(R_{q}\right)\cap S_{2}\left(1\right)}{∥\hat{\beta }-\beta^{*}∥}_{p}^{p}\geq c_{q,p}R_{q}{\left[\frac{\sigma_{z}^{2}\left(\sigma_{x}^{2}\sigma_{w}^{2}+\sigma_{z}^{2}\sigma_{e}^{2}\right)}{\sigma_{x}^{4}\kappa_{c}^{2}}\frac{\log d}{n}\right]}^{\frac{p-q}{2}}\right)\geq \frac{1}{2}.$$The proof is complete. □

Note that the probability 1/2 in Proposition 1 is just a standard convention, and it may be made arbitrarily close to 2/3 by choosing the universal constants suitably. Specifically, noting from Equation (Equation 10) that as long as $N_{2}\left(\epsilon \right)\geq 2$ is sufficiently large, the probability can be made sufficiently close to 2/3. The requirement on the sufficiently large value of $N_{2}\left(\epsilon \right)$ can be satisfied by choosing the universal constants $L_{q,2}$ and *c* in view of Equation (Equation 11).

**Proposition** **2.**
*In the additive error setting, suppose that for a universal constant $c_{1}$, $\hat{\Gamma }$ satisfies Assumption 2 with $\kappa_{l}>0$ and $\tau_{l}\left(n,d\right)\leq c_{1}R_{q}{\left(\frac{\log d}{n}\right)}^{1-q/2}$. Then there exist universal constants $\left(c_{2},c_{3}\right)$ and a constant $c_{q}$ depending only on q such that, with probability at least $1-c_{2}\exp\left(-c_{3}\log d\right)$, the minimax $\ell_{2}$-loss over the $\ell_{q}$-ball is upper bounded as*
(13)$$\min_{\hat{\beta }}\max_{\beta^{*}\in B_{q}\left(R_{q}\right)\cap S_{2}\left(1\right)}{∥\hat{\beta }-\beta^{*}∥}_{2}^{2}\leq c_{q}\left[\frac{\sigma_{z}^{2-q}{\left(\sigma_{w}+\sigma_{e}\right)}^{2-q}+\kappa_{l}^{1-q}}{\kappa_{l}^{2-q}}\right]R_{q}{\left(\frac{\log d}{n}\right)}^{1-q/2}.$$


**Proof.** It suffices to find an estimator for $\beta^{*}$, which has a small $\ell_{2}$-norm estimation error with high probability. We consider the estimator formulated as follows
(14)$$\hat{\beta }\in \underset{\beta \in B_{q}\left(R_{q}\right)\cap S_{2}\left(1\right)}{arg min}\left\{\frac{1}{2}\beta^{⊤}\hat{\Gamma }\beta -{\hat{\Upsilon }}^{⊤}\beta \right\}.$$It is worth noting that (Equation 14) involves solving a nonconvex optimization problem when q∈[0,1), while a near-global solution can be obtained efficiently by the algorithm proposed in [14]. Since $\beta^{*}\in B_{q}\left(R_{q}\right)\cap S_{2}\left(1\right)$, it follows from the optimality of $\hat{\beta }$ that $1/2{\hat{\beta }}^{⊤}\hat{\Gamma }\hat{\beta }-{\hat{\Upsilon }}^{⊤}\hat{\beta }\leq 1/2{\beta^{*}}^{⊤}\hat{\Gamma }\beta^{*}-{\hat{\Upsilon }}^{⊤}\beta^{*}$. Define $\hat{\Delta }:=\hat{\beta }-\beta^{*}$, and thus one has that $\hat{\Delta }\in B_{q}\left(2R_{q}\right)$. Then it follows that
$${\hat{\Delta }}^{⊤}\hat{\Gamma }\hat{\Delta }\leq 2\langle \Delta ,\hat{\Upsilon }-\hat{\Gamma }\beta^{*}\rangle .$$This inequality, together with the assumption that $\hat{\Gamma }$ satisfies Assumption 2, implies that
(15)$$\kappa_{l}∥\hat{\Delta }{∥}_{2}^{2}-\tau_{l}\left(R_{q},n,d\right)\leq 2\langle \hat{\Delta },\hat{\Upsilon }-\hat{\Gamma }\beta^{*}\rangle \leq 2∥\hat{\Delta }{∥}_{1}{∥\hat{\Upsilon }-\hat{\Gamma }\beta^{*}∥}_{\infty }.$$It then follows from [11] [Lemma 2] that there exist universal constants $\left(c_{2},c_{3},c_{4}\right)$ such that, with probability at least $1-c_{2}\exp\left(-c_{3}\log d\right)$,
(16)$$∥\hat{\Upsilon }-\hat{\Gamma }\beta^{*}{∥}_{\infty }\leq c_{4}\sigma_{z}\left(\sigma_{w}+\sigma_{e}\right){∥\beta^{*}∥}_{2}\sqrt{\frac{\log d}{n}}=c_{4}\sigma_{z}\left(\sigma_{w}+\sigma_{e}\right)\sqrt{\frac{\log d}{n}}.$$Combining (Equation 15) and (Equation 16), one has that
$$\kappa_{l}∥\hat{\Delta }{∥}_{2}^{2}\leq 2c_{4}\sigma_{z}\left(\sigma_{w}+\sigma_{e}\right)\sqrt{\frac{\log d}{n}}{∥\hat{\Delta }∥}_{1}+\tau_{l}\left(R_{q},n,d\right).$$Introduce the shorthand $\sigma :=\sigma_{z}\left(\sigma_{w}+\sigma_{e}\right)$. Recall that $\hat{\Delta }\in B_{q}\left(2R_{q}\right)$. It then follows from [24] [Lemma 5] (with $\tau =\frac{2c_{4}\sigma }{\kappa_{l}}\sqrt{\frac{\log d}{n}}$) and the assumption $\tau_{l}\left(R_{q},n,d\right)\leq c_{1}R_{q}{\left(\frac{\log d}{n}\right)}^{1-q/2}$ that
$$∥\hat{\Delta }{∥}_{2}^{2}\leq \sqrt{2R_{q}}{\left(\frac{2c_{4}\sigma }{\kappa_{l}}\sqrt{\frac{\log d}{n}}\right)}^{1-q/2}{∥\hat{\Delta }∥}_{2}+2R_{q}{\left(\frac{2c_{4}\sigma }{\kappa_{l}}\sqrt{\frac{\log d}{n}}\right)}^{2-q}+\frac{c_{1}}{\kappa_{l}}R_{q}{\left(\frac{\log d}{n}\right)}^{1-q/2}.$$Therefore, by solving this inequality with the indeterminate viewed as $∥\hat{\Delta }{∥}_{2}$, we arrive at the conclusion that there exists a constant $c_{q}$ depending only on *q* such that (Equation 13) holds with probability at least $1-c_{2}\exp\left(-c_{3}\log d\right)$. The proof is complete. □

**Remark 2.** (i) *The lower and upper bounds on minimax risks are dependent on the triple $\left(R_{q},n,d\right)$, the error level, and structural properties of the observed matrix Z, as shown in Propositions 1 and 2. Specifically, by setting p=2 in Proposition 1, the lower and upper bounds agree up to constant factors independent of the triple $\left(R_{q},n,d\right)$, showing the optimal minimax rate in the additive error case.*(ii) *Note that when p=2 and q=0 (i.e., the exact sparse case), the minimax rate scales as $\Theta \left(R_{0}\frac{\log d}{n}\right)$. In the high-dimensional regime when $d/R_{0}∼d^{\gamma }$ for some constant γ>0, this rate is equivalent to $R_{0}\frac{\log(d/R_{0})}{n}$ (up to constant factors), which re-captures the same scaling as in [19].*(iii) *The assumption that $\tau_{l}\left(R_{q},n,d\right)\leq c_{1}R_{q}{\left(\frac{\log d}{n}\right)}^{1-q/2}$ in Proposition 2 is not unreasonable. It has been shown in [11] [Lemma 1] that it can be satisfied with high probability for the high-dimensional linear errors-in-variables model.*

The following two theorems are on probabilistic consequences in view of conditions to ensure Assumptions 1 and 2. The proofs are obtained by applying Propositions 1 and 2 together with Remark 1(ii),(iii), respectively, as well as the elementary probability theory.

**Theorem** **1** (Lower bound on $\ell_{p}$-loss). *In the additive error setting, for any p∈[1,+∞), there exist universal positive constants $\left(c_{0},c_{1}\right)$ and a constant $c_{q,p}$ depending only on q and p such that, with probability at least $1/2(1-c_{1}\exp\left(-logd\right))$, the minimax $\ell_{p}$-loss over the $\ell_{q}$-ball is lower bounded as*
(17)$$\min_{\hat{\beta }}\max_{\beta^{*}\in B_{q}\left(R_{q}\right)\cap S_{2}\left(1\right)}{∥\hat{\beta }-\beta^{*}∥}_{p}^{p}\geq c_{q,p}{\left[\frac{\sigma_{x}^{2}\sigma_{w}^{2}+\sigma_{z}^{2}\sigma_{e}^{2}}{c_{0}^{2}\sigma_{x}^{4}}\right]}^{\frac{p-q}{2}}R_{q}.$$

**Proof.** (Equation 17) follows from substituting $\kappa_{c}=c_{0}\sigma_{z}\sqrt{\frac{\log d}{n}}$ for a universal positive constant $c_{0}$ to (Equation 4). As for the probability, we define the following two events A={Assumption 1 happens} and B={(Equation 17) happens}. Then Proposition 1 is applicable to conclude that P(B|A)≥1/2. Note from Remark 1(ii) that $P\left(A\right)\geq 1-c_{1}\exp\left(-logd\right)$ for a universal positive constant $c_{1}$. Then it follows from the elementary probability that $P\left(B\right)=P\left(B|A\right)P\left(A\right)+P\left(B|A^{c}\right)P\left(A^{c}\right)\geq P\left(B|A\right)P\left(A\right)\geq 1/2\left(1-c_{1}\exp\left(-logd\right)\right)$, which completes the proof. □

**Theorem** **2** (Upper bound on $\ell_{2}$-loss)**.**
*In the additive error setting, for a universal constant $c_{0}$, suppose that $\sigma_{x}^{2}\max\left(\frac{\sigma_{z}^{4}}{\sigma_{x}^{4}},1\right)\leq c_{0}R_{q}{\left(\frac{\log d}{n}\right)}^{-q/2}$. Then there exist universal constants $\left(c_{1},c_{2},c_{3},c_{4}\right)$ and a constant $c_{q}$ depending only on q such that, with probability at least $\left(1-c_{1}\exp\left(-c_{2}n\min\left(\frac{\sigma_{x}^{4}}{\sigma_{z}^{4}},1\right)\right)\right)\left(1-c_{3}\exp\left(-c_{4}\log d\right)\right)$, the minimax $\ell_{2}$-loss over the $\ell_{q}$-ball is upper bounded as*
(18)$$\min_{\hat{\beta }}\max_{\beta^{*}\in B_{q}\left(R_{q}\right)\cap S_{2}\left(1\right)}{∥\hat{\beta }-\beta^{*}∥}_{2}^{2}\leq c_{q}\left[\frac{\sigma_{z}^{2-q}{\left(\sigma_{w}+\sigma_{e}\right)}^{2-q}+\sigma_{x}^{2-2q}}{\sigma_{x}^{4-2q}}\right]R_{q}{\left(\frac{\log d}{n}\right)}^{1-q/2}.$$

**Proof.** (Equation 17) follows from substituting $\kappa_{l}=\frac{\sigma_{x}^{2}}{2}$ to (Equation 13). As for the probability, we define the following two events C={Assumption 2 happens} and D={(Equation 18) happens}. Then Proposition 2 is applicable to conclude that $P\left(D|C\right)\geq 1-c_{3}\exp\left(-c_{4}\log d\right)$ for universal positive constants $\left(c_{3},c_{4}\right)$. Note from Remark 1(iii) that $P\left(C\right)\geq 1-c_{1}\exp\left(-c_{2}n\min\left(\frac{\sigma_{x}^{4}}{\sigma_{z}^{4}},1\right)\right)$ for universal positive constant $\left(c_{1},c_{2}\right)$. Then it follows from the elementary probability that $P\left(D\right)=P\left(D|C\right)P\left(C\right)+P\left(D|C^{c}\right)P\left(C^{c}\right)\geq P\left(D|C\right)P\left(C\right)\geq \left(1-c_{1}\exp\left(-c_{2}n\min\left(\frac{\sigma_{x}^{4}}{\sigma_{z}^{4}},1\right)\right)\right)$ $\left(1-c_{3}\exp\left(-c_{4}\log d\right)\right)$, which completes the proof. □

## 4. Conclusions

We focused on the information-theoretic limitations of estimation for sparse linear regression with additive measurement errors under the high-dimensional scaling. The minimax rates of convergence were analyzed by virtue of lower and upper bounds for $\ell_{p}$-losses over $\ell_{q}$-balls based on information theory. The derived lower and upper bounds together revealed the influence of corruption in the observed covariates on parameter estimation. Note that the assumed Gaussian random design matrices are of particular interest and widely applied in the field where the design matrix can be chosen by researchers, such as compressed sensing and signal processing [32]. However, the independent Gaussian assumption is still somewhat restrictive, while our earlier work [14] provides upper bounds on estimation to sub-Gaussian matrices with nondiagonal covariances. Further research may generalize the current result, especially the estimation on lower bounds, to sub-Gaussian matrices with nondiagonal covariances or other types of measurement errors, such as the multiplicative errors or errors with dependent structures. In addition, due to the modern high-dimensional challenge, it is of great significance for a method to be adaptive in learning problems. Hence, it would be a prospective direction to analyze minimax optimal rates of convergence without knowledge of the sparsity degree, such as *q* and the $\ell_{q}$-radius.

## Data Availability

Not applicable.

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
