# Peer review of "Minimax Rates of ℓp-Losses for High-Dimensional Linear Errors-in-Variables Models over ℓq-Balls"

_entropy, 2021, doi:10.3390/e23060722_

Round 1
Reviewer 1 Report
The mathematical derivations in this paper appear to be quite sound. To the best of my knowledge, the specific problem studied in the paper has not appeared in previous literature. However, the results are a rather straightforward combination of some subset of ideas in references [15] and [20]. Thus, I feel that the novelty of the contributions is quite incremental.
Author Response
Thank you for your comment.
Reviewer 2 Report
This paper considers the high-dimensional linear regression model assuming weak sparsity of the regressor vector parameter, investigating minimax rates of convergence. The paper is short and well written, I could only find some minor errors which follow. My main query (below) relates to the relevance of the results given the assumptions employed.
Main query: How realistic is the assumption that the random predictor vector is Gaussian and that the mean is zero with a independence between the predictor variables? I see that in the Conclusion section there is mention of further research to generalize the results, but without these generalizations do the results have practical relevance. If so, I think it would be important to stress that. Also, the variance matrix for W is assumed known, and that too provides further limitations.
Abstract, line 4: "The by the" should be "Then by the"
Line 37: "researches" should be "researchers"
Line 106: it is stated that Wi ∈ Rn. Correct to Wi ∈ Rd.
Line 108: it is stated that ei is a vector with covariance matrix σ2eIn. However, ei is the ith element of the vector e. Change ei to e.
Is the estimator of Γ (page 3) guaranteed to be positive definite when the amount of noise is large comparable to variation in the the Xs?
Reviewer 3 Report
The minimax rates of lp-losses of convergence for estimation are investigated. They analyze their problem by following [23] and [11], but the sparsity assumption is generalized, which is a novel point of this paper. I think this paper is worth publishing, but it is better to be explained the following points before publishing.
Main concern:
- By taking the limit where q -> 0, do the results obtained here recover the existing results in the previous studies?
- In the end of the proof of Thm.1, the authors explain that Thm.1 can be extended to the case where the probability that the inequality in Thm.1 holds can be arbitrarily close to one. I think it is useful to explain its result explicitly.
Minor points:
- p.1, l.4: "The by the" -> "By".
- p.1, l.23: Delete Z_i. It's not defined yet.
- p.3, l.96: Delete \delta. It's a scalar.
- p.5, 1st line in proof of Thm.1: Define \delta here. It will be explained after Eq.(7).
- p.6, 1st line: Define \epsilon here. It will be explained after Eq.(7),
- I feel the authors should survey and cite recent papers concerning their problem setting if exist.
Reviewer 4 Report
The paper considers a linear regression model with the covariates corrupted by Gaussian noise, as well as traditional AWGN in observations. Under the soft sparsity of parameters based on the l_q norm for 0<=q<1, lower and upper bounds on the minimax l_p loss are derived. The derived bounds are tight when the l_2 loss is considered.
The main contribution is an extension of the exact sparsity q=0 in [15] to the general case 0<=q<1, as well as a generalization of the soft sparsity for clean data in [20-22] to for corrupted data. The authors follow a proof strategy in [24, Theorem 1] to prove Theorem 1 on the lower bound. The difference between [24] and the submitted paper is in how to select the pair (\delta, \epsilon) below (7). In proving Theorem 2 on the upper bound, the authors use existing technical results, such as [11, Lemma 2] and [20, Lemma 5].
The proof strategy for Theorems 1 and 2 is an incremental generalization of existing results. However, it is worth conducting the existing results and providing a new theoretical result.
Major Comment:
--My main concern is in Assumptions 1 and 2. Lemma 1 is essential in proving Theorem 1, so that X_{i.}, W_{i.}, and e_i must be jointly Gaussian. In this case, there are obviously some events such that Assumption 1 or 2 does not hold, as long as the system size is finite. Thus, Assumptions 1 and 2 have to be asymptotic statements in some limits with respect to n and d.
Specify limits with respect to n and d such that both Assumptions 1 and 2 hold with probability 1 or in the sense of the convergence in probability. Prove Theorems 1 and 2 under such modified assumptions. The standard union bounds should work by separating the events into two events: The conditions in Assumptions 1 and 2 hold in first events while they do not in the other events.
Other minor comments:
--Clarify whether the vector \beta^{*} is a column or row. In terms of the inner product in (1), it should be a row vector since X_{i.} is a row vector. In terms of \hat{Y} in line 109, it should be a column vector since \Sigma_{x}\beta^{*} must be well defined. My suggestion is to replace the inner product in (1) with X_{i.}\beta^{*}.
--line 16
Clarify the dimension of X_{i.}, i.e. X_{i.}\in\mathbb{R}^{1\times d}.
--line 106
The dimension of W_i should be 1\times d.
--line 117
Replace R^{n}\times R^{n\times d} with R^{n\times d}\times R^{d}.
--Assumption 2
The authors should need to impose some conditions on the function \tau_{l}. For instance, letting \tau_{l}=\infty imposes no restrictions on the matrix \hat{\Gamma}. My recommendation is to present the upper bound on \tau_{l} in Theorem 2 here.
--\sigma_x
\sigma_x>0 should be assumed clearly. We expect that the sample covariance \hat{\Gamma} reduces to \sigma_{x}^{2}I, which must be strictly positive-definite in terms of Assumption 2.
--Eq.(2)
The mean is not scalar since Z_{i.} is a d-dimensional row vector. Fix the typo.
Round 2
Reviewer 2 Report
I am satisfied with the comments regarding limitations included in the new version of the manuscript.
Author Response
Thank you for your comment.
Reviewer 3 Report
I think that the manuscript was properly revised. However, I want to confirm the statement in line 175-177. When N_2(\epsilon) is sufficiently large, the probability of (10) approaches 2/3. I think the authors should revise this point before publication.
Author Response
Thank you for your helpful comment. We have corrected the the probability of (10) as suggested in the revised manuscript; please see line 184-185.
Reviewer 4 Report
My major comments have not been addressed fully while minor and editorial comments have been addressed. The authors have added remarks below Assumptions 1 and 2 to clarify when the assumptions hold. However, they have not modified Assumptions 1, 2, or the proofs of Theorems 1 and 2. As a result, the proofs in the theorems still have to be improved.
Major comments:
--Assumption 1 should be probabilistic. Assume the last statement in Remark 1(ii) into Assumption 1, "there exist universal positive constants (c0, c1) that ...." Let E denote an event in which the upper bound in Assumption 1 holds. In the proof of Theorem 1, the probability on the LHS of (5) should be decomposed into two terms, i.e. P(min max ...) = P(E)P(min max ...|E) + P(E^c)P(min max ...|E^c). In evaluating a lower bound, the second term can be ignored. In terms of the first term, the probability P(E) may contribute to the final result while P(min max ...|E) have already bounded in the current proof.
--Assumption 2 should be also probabilistic. Assume \tau_l <= C(lod d/n)^{1-q/2} in Theorem 2 into Assumption 2, as well as adding a probabilistic condition, like "with probability at least ..." Since this upper bound is stronger than in the last statement in Remark 2(iii), with the exception of the case log d/n>1, the current proof of Theorem 2 might work without major changes. However, the authors should be careful with whether this change in Assumption 2 produces some gaps in the proof of Theorem 2.
